# Loopy: Taming Audio-Driven Portrait Avatar with Long-Term motion Dependency

**Jianwen Jiang**[1*†]**, Chao Liang**[1*]**, Jiaqi Yang**[1*]**, Gaojie Lin**[1]**, Tianyun Zhong**[2‡]**, Yanbo Zheng**[1]

[1]ByteDance, [2]Zhejiang University

{jianwen.alan,liangchao.0412,yjq850207131}@gmail.com
zhongtianyun@zju.edu.cn

## Abstract

With the introduction of the video diffusion model, audio-conditioned human video generation has recently achieved significant breakthroughs in both the naturalness of motion and the synthesis of portrait details. Due to the limited control of audio signals in driving human motion, existing methods often add auxiliary spatial signals, such as movement regions, to stabilize movements. However, this compromises the naturalness and freedom of motion. To address this issue, we propose an end-to-end audio-only conditioned video diffusion model named **Loopy**. Specifically, we designed two key modules: an inter- and intra-clip temporal module and an audio-to-latents module. These enable the model to better utilize long-term motion dependencies and establish a stronger audio-portrait movement correlation. Consequently, the model can generate more natural and stable portrait videos with subtle facial expressions, without the need for manually setting movement constraints. Extensive experiments show that Loopy outperforms recent audio-driven portrait diffusion models, delivering more lifelike and high-quality results across various scenarios. Video samples are available at this URL.

## 1 Introduction

Due to the rapid advancements of GAN and diffusion models in the field of video synthesis (Bar-Tal et al., 2024; Blattmann et al., 2023a;b; Guo et al., 2023; Zhou et al., 2022; Gupta et al., 2023; Wang et al., 2023; Ho et al., 2022; Brooks et al., 2022; Wang et al., 2020; Singer et al., 2022; Li et al., 2018; Villegas et al., 2022), human video synthesis (Siarohin et al., 2019; 2021; Xu et al., 2024b; Hu, 2024; Corona et al., 2024; Lin et al., 2024) has gradually approached the threshold of practical usability in terms of quality, attracting significant attention in recent years. Among these, zero-shot audio-driven portrait synthesis has seen an explosion of research (He et al., 2023; Tian et al., 2024; Xu et al., 2024a; Wang et al., 2024; Chen et al., 2024; Xu et al., 2024b; Stypulkowski et al., 2024) since around 2020, due to its ability to generate portrait videos with a low barrier to entry. Recently, diffusion model techniques have been introduced, with end-to-end audio-driven models (Tian et al., 2024; Xu et al., 2024a; Chen et al., 2024) demonstrating more vivid synthesis results compared to existing methods.

However, due to the weak correlation between audio and portrait motion, end-to-end audio-driven methods typically introduce additional spatial conditions to restrict movement areas and ensure temporal stability in the synthesized videos. While the introduction of preset motion templates for spatial conditions may mitigate this issue, it also introduces several problems related to template selection, audio-template synchronization and repetitive movements. These spatial conditions, or auxiliary designs, such as face locators and speed layers (Tian et al., 2024; Xu et al., 2024a; Chen et al., 2024), restrict the range and velocity of portrait movements while also hindering the full potential of video diffusion models in generating vivid motion. This is because the model tends to follow given movement information during training rather than learning to generate natural movements from audio. As shown in Figure 1, the overall richness of portrait motion in existing methods is limited. This

---

[*]Equal Contribution

[†]Project Lead

[‡]Done during an internship at ByteDance.

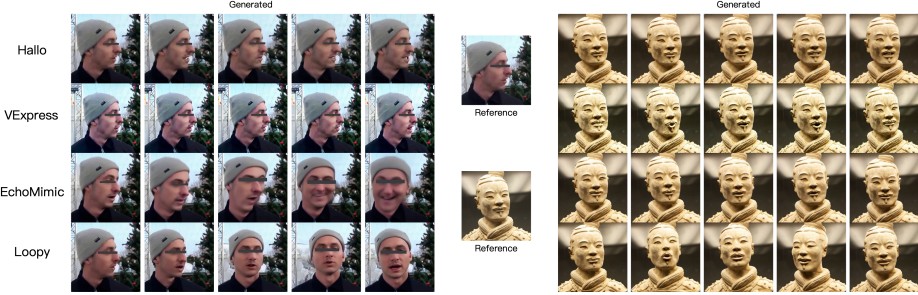

Figure 1: **Visual comparisons with existing methods.** Existing methods struggle to generate natural movements. Compared to reference images, their motion, posture, and expressions often resemble the reference or remain nearly static due to the auxiliary spatial conditions. In contrast, Loopy effectively generates natural movements solely from audio, including detailed head movements and facial expressions. Video samples are provided in the supplementary materials.

paper aims to address this issue by proposing an audio-only conditioned portrait diffusion model, enabling the model to learn natural motion patterns entirely from data without the need for spatial constraints.

In the absence of spatial conditions, the input information is insufficient to accurately infer portrait movement trends, which leads to inconsistent and frequently changing motion patterns. We discovered that employing the CFG technique (Ho & Salimans, 2022) can stabilize videos by making them relatively static in comparison to the reference image. However, this approach is suboptimal because it fails to teach the model natural motion generation. It merely replicates expressions and poses from the reference image, ultimately resulting in a decline in overall quality. In diffusion-based frameworks, motion is influenced by both audio and frames from preceding clips, referred to as motion frames. Motion frames provide appearance information from preceding clips, which strongly influences motion generation. However, current methods typically use fewer than 5 motion frames, covering only about 0.2 seconds at 25 FPS. This brief duration causes the model to extract appearance information rather than temporal information, such as motion style. For instance, 0.2 seconds of preceding information is insufficient for the model to determine whether eye blinking should occur, making it a random event rather than a naturally generated expression. When motion style is difficult to determine from audio and motion frames, it exhibits randomness, necessitating additional guidance from spatial conditions such as face boxes and movement speed. We attempted to directly increase the length of motion frames and found that this can generate more dynamic facial movement details, although it is not stable. This suggests that appropriately increasing the temporal receptive field may help capture motion patterns more effectively and aid in generating natural motion.

Besides limited expressiveness caused by restricted movement, current methods also lack dynamic detail in facial movements, resulting in stiff and unnatural expressions beyond the mouth area, as shown in Figure 1. This issue may stem from the weak correlation between audio and portrait motion, which causes the model to focus more on modeling the relationship between audio and video pixels, including a significant amount of irrelevant background motion, rather than the relationship between audio and portrait motion. This phenomenon has also been noted in some studies (Xu et al., 2024a).

Building on the above observations and considerations, we propose an end-to-end audio-conditioned diffusion model for portrait video generation, Loopy, leveraging long-term motion dependencies to generate vivid portrait videos. Specifically, in terms of the temporal aspect, we designed inter- and intra-clip temporal modules. Motion frames are modeled with a separate temporal layer to capture inter-clip relationships, whereas the original temporal module focuses on intra-clip modeling. Additionally, inter-clip temporal layers are equipped with a temporal segment module that extends the receptive field to over 100 frames (approximately 5 seconds at 25 fps, 30 times the original). These modules help the model better leverage long-term motion dependency information, allowing it to learn natural and stable motion patterns from data without the need for movement constraints. Furthermore, in terms of the audio aspect, we introduced the audio-to-latents module, which transforms audio and facial motion-related features (such as head movement variance and expression variance) into motion latents in a shared feature space. These latents are then inserted into the denoising network as conditions. During testing, motion latents are generated solely from audio. This approach allows weakly motion-correlated audio to leverage strongly correlated conditions, thereby enhancing the

relationship between audio and portrait motion, and improving subtle facial expressions. Extensive experiments validate that our design effectively enhances both the naturalness of motion and the robustness of video synthesis across various types of input images and audio combinations. In summary, our contributions are as follows:

(1) We propose Loopy, an audio-driven diffusion model for portrait video generation. It features two key components: inter- and intra-clip temporal modules designed to learn natural motion patterns from long-term dependencies, and an audio-to-latents module that enhances audio-portrait motion correlation by using strongly correlated conditions during training. Loopy is capable of generating vivid talking portrait videos with subtle facial details without relying on motion constraints or templates.

(2) We validated the effectiveness of our method on public datasets and evaluated the model's capabilities across various scenarios, including different types of input images and audio. The results demonstrate that Loopy achieves more lifelike and stable synthesis compared to existing methods.

## 2 RELATED WORKS

Audio-driven portrait video generation has attracted significant attention in recent years, with numerous works advancing the field. Most of these methods can be categorized into GAN-based and diffusion-based approaches according to their video synthesis techniques.

GAN-based methods (Zhou et al., 2020; Prajwal et al., 2020; Zhang et al., 2023b; Liang et al., 2022; Jiang et al., 2024) typically consist of two key components: an audio-to-motion model and a motion-to-video model. These models are usually implemented independently. For example, MakeItTalk (Zhou et al., 2020) uses an LSTM module to predict landmarks based on the input audio, and then a warp-based GAN model converts the landmarks into video. SadTalker (Zhang et al., 2023b) utilizes the existing FaceVid2Vid (Wang et al., 2021) method as the image synthesizer, employing ExpNet and PoseVAE to transform audio features into the inputs required by FaceVid2Vid, thereby completing the audio-to-video generation. With the introduction of diffusion techniques, some methods have implemented the audio-to-motion module using diffusion models while retaining the independent implementation of the motion-to-video module. For instance, GAIA (He et al., 2023) uses a VAE to represent motion as motion latents and implements a motion latents-to-video generation model. Furthermore, it designs a diffusion model to achieve audio-to-motion latents generation, thereby enabling audio-to-video generation. DreamTalk (Ma et al., 2023), Dream-Talk (Zhang et al., 2023a), and VASA-1 (Xu et al., 2024b) propose similar ideas, using PIRender (Ren et al., 2021), FaceVid2Vid (Wang et al., 2021), and MegaPortrait (Drobyshev et al., 2022) as their motion-to-video models, respectively, and designing audio-to-motion diffusion models to complete the audio-to-portrait video generation process. Additionally, some methods Ye et al. (2024) also incorporate NeRF for video rendering, combined with an audio-to-motion model to achieve audio-driven portrait video generation.

Apart from the above types, DiffusedHead (Stypulkowski et al., 2024) and EMO (Tian et al., 2024) achieves audio-to-portrait video generation using a single diffusion model, replacing the two-stage independent design of the audio-to-motion module and the motion-to-video model. Recent methods, such as Hallo (Xu et al., 2024a), EchoMimic (Chen et al., 2024) and VExpress (Wang et al., 2024), improve the audio-to-video modeling capabilities by introducing techniques like attention reweighting and spatial loss based on the end-to-end audio-to-video diffusion framework but still retain motion constraint conditions. Although these end-to-end methods can generate decent portrait videos, they need to introduce spatial condition module, like face locator and speed layer, to constrain portrait movements for stability, limiting the model's ability to generate diverse motions in practical applications and hindering the full potential of diffusion models.

## 3 METHOD

In this section, we first provide an overview of the Loopy framework, including its input, output and key designs. Second, we focus on the design of the inter/intra- temporal modules, including the temporal segment module. Third, we detail the implementation of the audio-to-latents module. Finally, we describe the implementation details for training and testing Loopy.

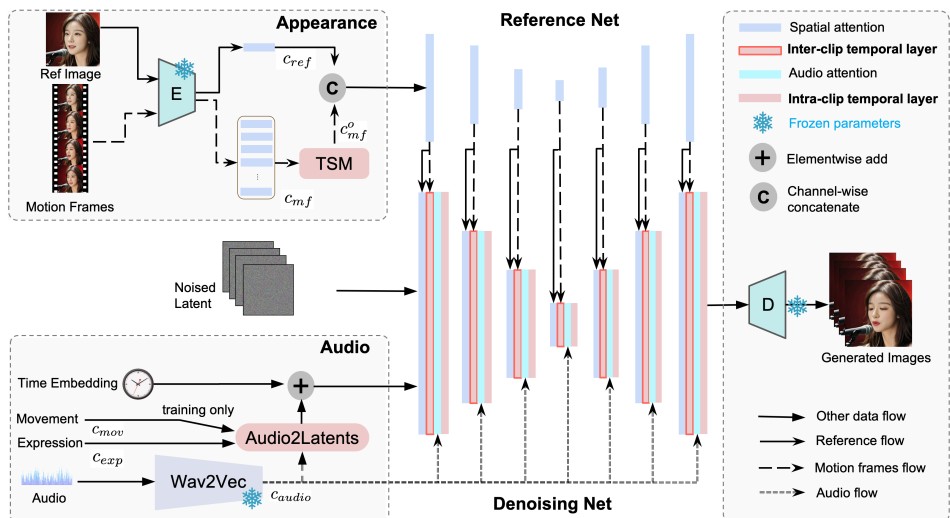

Figure 2: **The framework of Loopy.** it removes the commonly used face locator and speed layer modules in existing methods. Instead, it achieves flexible and natural motion generation through the proposed inter/intra-clip temporal layers, temporal segment module (TSM) and audio-to-latents modules.

## 3.1 FRAMEWORK

Our method is built upon Stable Diffusion (SD) and uses pretrained weights for initialization. SD is a text-to-image diffusion model based on the Latent Diffusion Model (LDM) (Rombach et al., 2022). It employs a pretrained VQ-VAE (Kingma, 2013; Van Den Oord et al., 2017) $\mathcal{E}$ to transform images from pixel space to latent space. During training, images are first converted to latents, i.e., $z_0 = \mathcal{E}(I)$. Gaussian noise $\epsilon$ is then added to the latents in the latent space based on the Denoising Diffusion Probabilistic Model (DDPM) (Ho et al., 2020) for $t$ steps, resulting in a noisy latent $z_t$. The denoising net takes $z_t$ as input and predicts $\epsilon$. The training objective can be formulated as follows.

$$L = \mathbb{E}_{z_t, c, \epsilon \sim \mathcal{N}(0,1), t} \left[ \| \epsilon - \epsilon_\theta(z_t, t, c) \|_2^2 \right], \tag{1}$$

where $\epsilon_\theta$ represents the denoising net, including condition-related modules, which is the main part Loopy aims to improve. $c$ represents the text condition embeddings in SD, and in Loopy, it is replaced by audio, motion frames, and other additional information influencing the final generation. During testing, the final image is obtained by sampling from gaussian noise and removing the noise based on DDIM (Song et al., 2020) or DDPM.

As demonstrated in Figure 2, in the Loopy, the inputs to the denoising net include noisy latents $z_t$. Unlike the original SD, where the input is a single image, here the input is a sequence of images representing a video. The inputs also include reference latents $c_{ref}$ (encoded reference image via VQ-VAE), audio embeddings $c_{audio}$ (audio features of the current clip), motion frames $c_{mf}$ (image latents of the frames from the preceding clips) and timestep $t$. During training, additional facial movement-related features are involved, the head movement variance $c_{mov}$ and the expression variance $c_{exp}$ of the current clip. The output is the predicted noise $\epsilon$. The denoising network employs a dual U-Net architecture (Hu, 2024; Zhu et al., 2023). This architecture includes an additional reference network, which replicates the original SD U-Net structure but utilizes reference latents $c_{ref}$ as input. The reference network operates concurrently with the denoising U-Net. During the spatial attention layer computation in the denoising U-Net, the key and value features from corresponding positions in the reference network are concatenated with the denoising U-Net's features along the spatial dimension before being processed by the attention module. This design enables the denoising U-Net to effectively incorporate reference image features from the reference network. Additionally, the reference network also takes motion frames latents $c_{mf}$ as input for feature extraction, allowing these features to be utilized in subsequent temporal attention computations.

## 3.2 INTER/INTRA- CLIP TEMPORAL MODULE

Here, we introduce the design of the proposed inter/intra- clip temporal modules. Unlike existing methods (Tian et al., 2024; Xu et al., 2024a; Chen et al., 2024; Wang et al., 2024) that process motion frame latents and noisy latent features simultaneously through a single temporal layer, Loopy employs two temporal attention layers, the inter-clip temporal layer and the intra-clip temporal layer. The inter-clip temporal layer first handles the cross-clip temporal relationships between motion frame latents and noisy latents, while the intra-clip temporal layer focuses on the temporal relationships within the noisy latents of the current clip.

**Inter/Intra- Temporal Lyaers.** First, we introduce the inter-clip temporal layer, initially ignoring the temporal segment module in Figure 2. As shown in Figure 3, we first collect the image latents from the preceding clip, referred to as motion frames latents. Similar to $c_{ref}$, these latents are processed frame-by-frame through the reference network for feature extraction. Within each residual block, the motion frames latent features obtained from the reference network are concatenated with the noise latent features from the denoising U-Net along the temporal dimension. To distinguish the types of latents, we add learnable temporal embeddings. Subsequently, self-attention is computed on the concatenated tokens along the temporal dimension, i.e., temporal attention. The intra-clip temporal layer differs in that its input does not include features from motion frames latents, it only processes features from the noisy latents of the current clip. By using inter/intra-clip temporal layers, the model better handles the aggregation of semantic temporal features across clips.

**Temporal Segment Module.** Due to the design of the inter-clip temporal layer, Loopy can better model the motion relationships among clips. To further enhance this capability, we introduce the temporal segment module before $c_{mf}$ enters the reference network. This module not only extends the temporal range covered by the inter-clip temporal layer but also extracts temporal information at different granularities based on the distance of each preceding clip from the current clip, as illustrated in Figure 3. The temporal segment module divides the original motion frame into multiple segments and extracts representative motion frames from each segment to abstract the segment. Based on these abstracted motion frames, we recombine them to obtain new motion frame latents, $c_{mf}^{o}$, for subsequent inter-clip temporal module computations. For the segmenta-

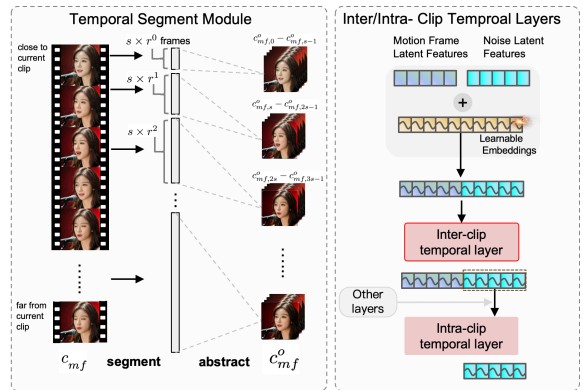

Figure 3: **The illustration of the temporal segment module and the inter/intra- clip temporal layers.** The former allows us to expand the motion frame to cover over 100 frames, while the later enables the modeling of long-term motion dependency.

tion process, we define two hyperparameters, stride $s$ and expand ratio $r$. The stride $s$ represents the number of abstract motion frames in each segment, while the expand ratio $r$ is used to calculate the number of original motion frames covered in each segment. The number of frames in the $i$-th segment, ordered from closest to farthest from the current clip, is given by $s \times r^{(i-1)}$. For example, with a stride $s = 4$ and an expand ratio $r = 2$ (also our default setting, and the total segment number is set to 5), the first segment would contain 4 frames, the second segment would contain 8 frames, and the third segment would contain 16 frames. For the abstraction process after segmentation, we default to uniform sampling within each segment. In the experimental section, we investigate different segmentation parameters and abstraction methods. Different approaches significantly impact the results, as segmentation and abstraction directly affect the learning of long-term motion dependencies. The output of the temporal segment module, $c_{mf}^{o}$, can be defined as:

$$c_{mf,i}^{o} = c_{mf,\left[\sum_{j=0}^{k-1} r^j \cdot s + r^k \cdot (i \mod s)\right]} \tag{2}$$

where $k = \left\lfloor \frac{i}{s} \right\rfloor$. $c_{mf,i}^{o}$ is the $i$-th element of the output and represents the $(i \mod s)$-th abstracted appearance information of the k-th segment.

The temporal segment module rapidly expands the temporal coverage of the motion frames input to the inter-clip temporal layer while maintaining acceptable computational complexity. For closer frames, a lower expansion rate retains more details, while for distant frames, a higher expansion rate covers a longer duration. This approach helps the model better capture motion style from long-term motion information and generate temporally natural motion without spatial constraints.

### 3.3 AUDIO CONDITION MODULE

For the audio condition, we first use wav2vec (Baevski et al., 2020; Schneider et al., 2019) for audio feature extraction. Following the approach in EMO (Tian et al., 2024), we concatenate the hidden states from each layer of the wav2vec network to obtain multi-scale audio features. For each video frame, we concatenate the audio features of the two preceding and two succeeding frames, resulting in a 5-frame audio feature as audio embeddings $c_{audio}$ for the current frame. Firstly, in each residual block, we use the commonly adopted cross-attention with noisy latents as the query and audio embeddings $c_{audio}$ as the key and value to compute an attended audio feature. This attended audio feature is then added to the noisy latent features obtained from inter-clip temporal layer, resulting in a new noisy latent features. This provides a preliminary audio condition for the model.

**Audio-to-Latents Module.** Additionally, as illustrated in Figure 4, we introduce audio-to-latents module to enhance the influence of audio on portrait motion. This module receives various inputs during training, including head movement and expression variances (strongly correlated with portrait motion, details are provided in the Appendix) and audio features (weakly correlated). The module maps these conditions to a shared motion latents space, which replaces the original conditions in subsequent computations.

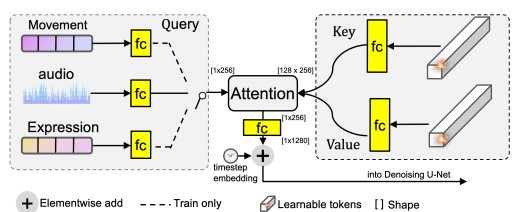

Figure 4: The audio-to-latents module.

This method allows audio to more accurately influence motion by leveraging the shared motion latents learned from strong conditions, also avoiding focus on irrelevant areas. Specifically, we maintain a set of learnable embeddings. For each input condition, we map it to query features using a fully connected (FC) layer, while the learnable embeddings serve as key and value features for attention computation to obtain a new feature based on the learnable embeddings. This attention computation is conducted on the token dimension. Audio and variance features are transformed into a query feature via different FC layers (token count is 1). For key and value features, the token count corresponds to the number of learnable embedding vectors, which we have set to 128. The FC layers also unify the QKV features to 256 channels for attention computation. The obtained value features, i.e., motion latents, are transformed to 1280 channels via an FC layer and added to the timestep embedding. During training, we sample an input condition for the audio-to-latents module with equal probability from audio embeddings, head absolute movement variance and face expression variance. During testing, we only input audio features to generate motion latents. In fact, expressions and movements can also influence motion generation through motion latents, as demonstrated on the project page.

### 3.4 TRAINING STRATEGIES

**Conditions Mask and Dropout.** In the Loopy framework, various conditions are involved, including the reference image $c_{ref}$, audio features $c_{audio}$, preceding frame motion frames $c_{mf}$, and motion latents representing audio and facial movement conditions. To better learn the unique information specific to each condition, we use distinct masking strategies for the conditions during the training process. During training, $c_{audio}$ and motion latents are masked to all-zero features with a 10% probability. For $c_{ref}$ and $c_{mf}$, we design specific dropout and mask strategies due to their highly overlapping information. $c_{mf}$ also provides appearance information and is closer to the current clip compared to $c_{ref}$, leading the model to heavily rely on motion frames rather than the reference image. To address this, $c_{ref}$ has a 15% probability of being dropped, meaning the denoising U-Net will not concatenate features from the reference network during spatial-attention computation. When $c_{ref}$ is dropped, motion frames are also dropped, meaning the denoising U-Net will not concatenate features from the reference network during temporal attention computation. Additionally, motion frames have an independent 40% probability of being masked to all-zero features.

**Multistage Training.** Following AnimateAnyone (Hu, 2024) and EMO (Tian et al., 2024), we employ a 2-stage training process. In the first stage, the model is trained without temporal layers and the audio condition module. The inputs to the model are the noisy latents of the target single-frame image and reference image latents, focusing the model on an image-level pose variations task. After completing the first stage, we proceed to the second stage, where the model is initialized with the reference network and denoising U-Net from the first stage. We then add the inter/intra- clip temporal modules and the audio condition module for full training to obtain the final model.

**Inference.** During Inference, we perform class-free guidance (Ho & Salimans, 2022) using multiple conditions. Specifically, we conduct three inference runs, differing in whether certain conditions are dropped. The final noise $e_{final}$ is computed as:

$$e_{final} = \text{audio\_ratio} \times (e_{audio} - e_{ref}) + \text{ref\_ratio} \times (e_{ref} - e_{base}) + e_{base}$$

where $e_{audio}$ includes all conditions, $e_{ref}$ drops the $c_{audio}$ condition, and $e_{base}$ further drops the $c_{ref}$. The audio ratio is set to 5 and the reference ratio to 3. We use DDIM with 25 denoising steps for inference. Ungenerated parts of motion frames are masked to all-zero features until generated.

## 3.5 Experiments

**Datasets.** For training data, we collected and filtered talking head videos from multiple sources, including face-related datasets, general-purpose datasets, and online platforms, excluding videos with audio that have low lip-sync scores (Chung & Zisserman, 2017), excessive head movement, and rotation. This resulted in 174 hours of training data. The dataset are detailed in the Appendix A. For test sets, we randomly sampled 100 videos from CelebV-HQ (Zhu et al., 2022) (a public high-quality celebrity video dataset with mixed scenes), RAVDESS (Kaggle) (a public high-definition indoor talking scene dataset with rich emotions) and HDTF (Zhang et al., 2021). To test the generalization ability of diffusion-based models, we also collected 20 portrait test images, including real people, anime, side face, and humanoid crafts of different materials, along with 20 audio clips, including speeches, singing, rap and emotionally rich speech. We refer to this test set as the openset test set.

**Implementation Details.** We trained our model using 24 Nvidia A100 GPUs with a batch size of 24, using the AdamW (Loshchilov & Hutter, 2017) optimizer with a learning rate of 1e-5 to train the model for two stages, each lasting 4 days. The generated video length was set to 12 frames, and the motion frame was set to 124 frames, representing the preceding 124 frames of the current 12-frame video. After temporal segment module, this was abstracted to 20 motion frame latents. During training, the reference image was randomly selected from frames within the video. For the facial motion information required to train audo-to-motion module, we used DWPose (Yang et al., 2023) to detect facial keypoints for the current 12 frames. The variance of the absolute position of the nose tip across these 12 frames was used as the absolute head movement variance. The variance of the displacement of the upper half of the facial keypoints (37 keypoints) relative to the nose tip across these 12 frames was used as the expression variance. The training videos were uniformly processed at 25 FPS and cropped to 512×512 portrait videos.

**Metrics and Compared Baselines.** We assess image quality using the IQA metric (Wu et al., 2023), video motion stability with VBench's smooth metric (Huang et al., 2024), and audio-visual synchronization with SyncC and SyncD (Chung & Zisserman, 2017). For the CelebvHQ and RAVDESS test sets, which have corresponding ground truth videos, we also compute FVD (Unterthiner et al., 2019), E-FID (Tian et al., 2024), and FID metrics for comparison. To demonstrate lifelike portrait movement beyond just lip movement, we provide global motion metrics (Glo) and dynamic expression metrics (Exp). DGlo and DExp represent the absolute differences from the ground truth, calculated based on the variance of key points of the nose and upper face, excluding the mouth area. More details are provided in the Appendix. For the openset test set, which lacks ground truth video references, we conducted subjective evaluations. Ten users assessed six dimensions: identity consistency, video synthesis quality, audio-emotion matching, motion diversity, naturalness of motion, and lip-sync accuracy. Participants identified the top-performing method in each dimension. For baseline methods, We compared recent state-of-the-art audio-driven portrait methods based on diffusion models, including Hallo (Xu et al., 2024a), VExpress (Wang et al., 2024) and EchoMimic (Chen et al., 2024), and also included the competitive GAN-based method, SadTalker Zhang et al. (2023b), as a reference.

Table 1: Comparisons with existing methods on the CelebV-HQ test set.

| Method | IQA↑ | Sync-C↑ | Sync-D↓ | FVD-R↓ | FVD-I↓ | FID↓ | Glo | Exp | DGlo↓ | DExp↓ | E-FID↓ |
|--------|------|---------|---------|--------|--------|------|-----|-----|-------|-------|--------|
| SadTalker | 2.953 | 3.843 | 8.765 | 171.848 | 1746.038 | 36.648 | 0.554 | 0.270 | 0.291 | 0.368 | 2.248 |
| Hallo | 3.505 | 4.130 | 9.079 | 53.992 | 742.974 | 35.961 | 0.499 | 0.255 | 0.301 | 0.329 | 2.426 |
| Hallo* | 3.467 | 2.607 | 10.875 | 63.142 | 883.249 | 48.676 | 0.489 | 0.412 | 0.313 | 0.420 | 2.516 |
| VExpress | 2.946 | 3.547 | 9.415 | 117.868 | 1356.510 | 65.098 | 0.020 | 0.166 | 0.339 | 0.464 | 2.414 |
| EchoMimic | 3.307 | 3.136 | 10.378 | 54.715 | 828.966 | 35.373 | 2.259 | 0.640 | 0.260 | 0.442 | 3.018 |
| Loopy | **3.780** | **4.849** | **8.196** | **49.153** | **680.634** | **33.204** | 2.233 | 0.452 | 0.279 | **0.309** | 2.307 |

Table 2: Comparisons with existing methods on the RAVDESS test set.

| Method | IQA↑ | Sync-C↑ | Sync-D↓ | FVD-R↓ | FVD-I↓ | FID↓ | Glo | Exp | DGlo↓ | DExp↓ | E-FID↓ |
|--------|------|---------|---------|--------|--------|------|-----|-----|-------|-------|--------|
| SadTalker | 3.840 | 4.304 | 7.621 | 22.516 | 487.924 | 32.343 | 0.604 | 0.120 | 0.271 | 0.213 | 3.270 |
| Hallo | 4.393 | 4.062 | 8.552 | 38.471 | 537.478 | 19.826 | 0.194 | 0.080 | 0.299 | 0.243 | 3.785 |
| Hallo* | 4.233 | 2.878 | 9.672 | 47.758 | 691.103 | 34.973 | 0.354 | 0.129 | 0.291 | 0.191 | 3.621 |
| VExpress | 3.690 | 5.001 | 7.710 | 62.388 | 982.810 | 26.736 | 0.007 | 0.039 | 0.329 | 0.283 | 3.901 |
| EchoMimic | 4.504 | 3.292 | 9.096 | 54.115 | 688.675 | 21.058 | 0.641 | 0.184 | 0.263 | 0.182 | 3.350 |
| Loopy | **4.506** | 4.814 | 7.798 | **16.134** | **394.288** | **17.017** | 2.962 | 0.343 | **0.260** | 0.197 | **3.132** |

We also included the results of the open-source method Hallo, trained on our dataset according to the official instructions, marked with *.

### 3.5.1 RESULTS AND ANALYSIS

**Performance in Complex Scenarios.** CelebV-HQ features videos of celebrities speaking in diverse scenarios, both indoors and outdoors, with various portrait poses. This makes testing on this dataset effectively simulate real-world usage conditions. As shown in Table 1, our method significantly outperforms the other compared methods in most metrics, as evidenced by the comparison videos provided in the supplementary materials. Regarding motion-related metrics, in the dynamic expression metric (Exp), our method closely matches GT, outperforming other methods. For global motion (Glo), our performance is similar to EchoMimic. However, our method distinctly excels in video synthesis quality and lip-sync accuracy. For Hallo trained on our training set (marked with *), the results did not improve, and similar outcomes in subsequent experiments indicate that the design of our method is key to achieving satisfactory results.

**Performance in Emotional Expression.** RAVDESS is a high-definition talking scene dataset containing videos with varying emotional intensities. It effectively evaluates the method's performance in emotional expression. The experimental results are listed in Table 2. As shown by the E-FID (Tian et al., 2024) metric, our method outperforms the compared methods. This is corroborated in the motion dynamics metrics Glo and Exp, where our results are close to the ground truth. Although our lip-sync accuracy is slightly inferior to VExpress, it is important to note that the results generated by VExpress generally lack dynamic motion, as indicated by the Glo, Exp, and E-FID metrics. This static nature can provide an advantage when measured with SyncNet for lip-sync accuracy.

**Performance in Openset Scenarios.** Compared to objective metrics, subjective evaluations more directly reflect the overall performance of the methods. We compared different input styles (real people, anime, humanoid crafts, and side faces) and various types of audio (speech, singing, rap, and emotional audio) to evaluate the performance of the methods. As shown in Figure 5, Loopy consistently and significantly outperforms the compared methods across these diverse scenarios.

### 3.5.2 ABLATION STUDIES

**Analysis of Key Components.** In this part, we conducted experiments on two datasets: openset and HDTF. The former includes complex and challenging inputs, while the latter features relatively simple scenes. We analyzed the impact of the two key components of Loopy, the inter/intra- clip temporal module and the audio-to-latents module. For the former, we conducted two experiments: (1) Removing the inter-clip temporal layer, similar to previous methods. (2) Removing the temporal segment module and retaining 4 motion frames as in other methods. For the latter, we removed the

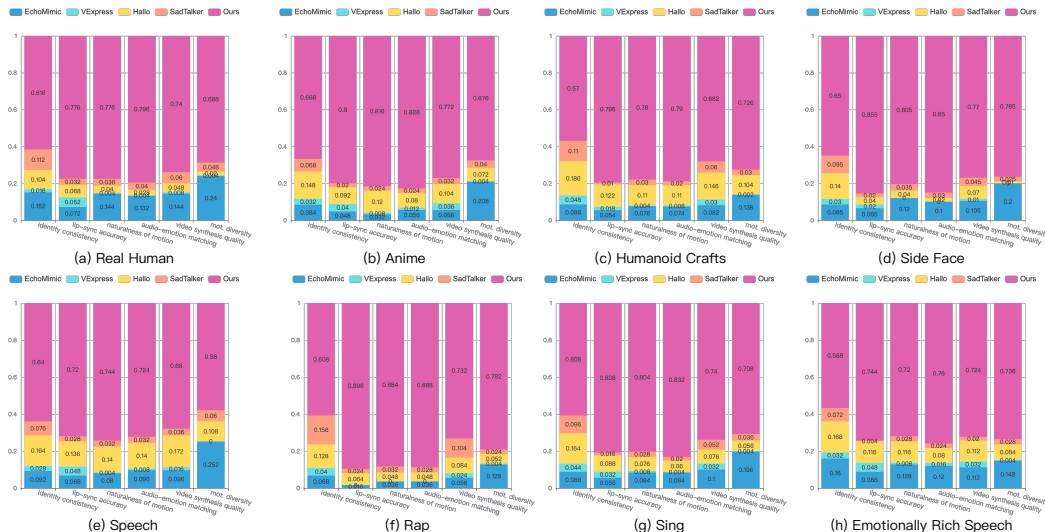

Figure 5: **User voting comparisons on the openset test set.** The first row presents results for different categories of input images, while the second row shows results for input audio.

Table 3: Comparisons and ablation studies on the openset and HDTF test sets.

| Method | Openset Test | | | | HDTF Test | | | | | |
|---|---|---|---|---|---|---|---|---|---|---|
| | IQA↑ | Sync-C↑ | Sync-D↓ | Smo. | IQA↑ | Sync-C↑ | Sync-D↓ | FVD-R↓ | FID↓ | E-FID↓ |
| Loopy (Full) | **4.507** | **6.303** | **7.749** | 0.9932 | **4.017** | **8.576** | 6.805 | **10.443** | **18.021** | 1.359 |
| Loopy (HDTF) | - | - | - | - | 3.894 | 8.176 | 7.097 | 12.742 | 21.733 | 1.476 |
| SadTalker | 3.749 | 5.390 | 9.586 | 0.9947 | 3.435 | 7.320 | 7.870 | 24.939 | 25.353 | 1.559 |
| EchoMimic | 4.447 | 3.674 | 10.494 | 0.9896 | 3.994 | 5.546 | 9.391 | 18.718 | 19.015 | 1.328 |
| Hallo | 4.412 | 5.483 | 8.798 | 0.9924 | 3.922 | 7.263 | 7.917 | 21.717 | 20.159 | 1.337 |
| Hallo* | 4.413 | 5.334 | 8.803 | 0.9937 | 3.846 | 3.579 | 11.151 | 29.419 | 24.689 | 1.543 |
| VExpress | 3.941 | 4.828 | 9.572 | 0.9961 | 3.482 | 8.186 | 7.382 | 48.038 | 30.916 | 1.506 |
| w/o inter-clip temp. | 4.335 | 6.104 | 8.129 | 0.9942 | 3.769 | 8.355 | 6.899 | 11.595 | 19.277 | 1.486 |
| w/o TSM | 4.386 | 6.054 | 8.235 | 0.9922 | 3.908 | 7.920 | 7.461 | 11.552 | 19.023 | 1.401 |
| w/o A2L | 4.428 | 5.999 | 8.351 | 0.9922 | 3.918 | 8.127 | 7.188 | 10.837 | 18.153 | 1.383 |
| 1 temp.+ 20 MF | 4.072 | 6.201 | 8.309 | 0.9752 | 3.765 | 8.494 | 6.795 | 11.817 | 19.819 | 1.380 |
| $s=1$, $r=2$ | 4.461 | 5.919 | 8.245 | 0.9940 | 3.945 | 7.848 | 7.388 | 14.814 | 19.665 | 1.392 |
| $s=2$, $r=2$ | 4.453 | 5.855 | 8.326 | 0.9930 | 3.946 | 8.172 | 7.038 | 13.310 | 18.690 | 1.398 |
| $s=3$, $r=2$ | 4.443 | 6.083 | 8.161 | 0.9930 | 3.941 | 7.985 | 7.263 | 11.260 | 18.655 | 1.322 |
| $s=4$, $r=1$ | 4.424 | 6.219 | 8.004 | 0.9931 | 3.937 | 8.419 | 6.877 | 10.672 | 18.210 | 1.299 |
| mean sample | 4.452 | 5.907 | 8.199 | 0.9931 | 3.865 | 7.851 | 7.570 | 10.506 | 19.475 | 1.535 |
| random sample | 4.438 | 6.098 | 8.144 | 0.9932 | 3.865 | 7.866 | 7.588 | 10.824 | 18.229 | 1.440 |

audio-to-latents module and used only cross-attention for audio feature injection. The results, listed in Table 3, indicate that removing the dual temporal layer design, the temporal segment module and the audio-to-latents module all lead to a decline in overall synthesis quality, with the impact being particularly significant when the dual temporal layer design is removed. The Table 3 also includes experimental results of comparison methods on these two datasets and Loopy's performance on the HDTF test set when trained without collected data for reference. These results substantiate the efficacy of our proposed modules.

**Impact of Long-Term Temporal Dependency.** We further investigated the impact of long-term motion dependency on the model and listed the results in Table 3. Initially, we compared the effects of extending the motion frame length to 20 under a single temporal layer setting. While this approach enhanced the model's dynamics, it significantly degraded overall quality (evidenced by a low smooth value). In contrast, the full model with 20 motion frames ($s=4$, $r=1$) and inter/intra-clip temporal layers improved stability, especially on the openset test set. This indicates that when the motion frame length exceeds the current clip, the dual temporal layer design better utilizes long-term motion

information, although improper handling may have negative effects. Regarding long-term motion coverage, we fixed $r=2$ and found that smaller $s$ values performed worse. Combined with other ablation results, this suggests that fewer motion frames contain less motion information, leading the network to extract more appearance information. This validates the effectiveness of long-term motion information in enhancing synthesis results. Additionally, we compared different abstraction strategies for motion frames within the temporal segment module. The default uniform sampling approach proved more effective than average pooling and random sampling, likely because it provides more stable and clearer interval information, aiding the inter-clip temporal layer in learning long-term motion information.

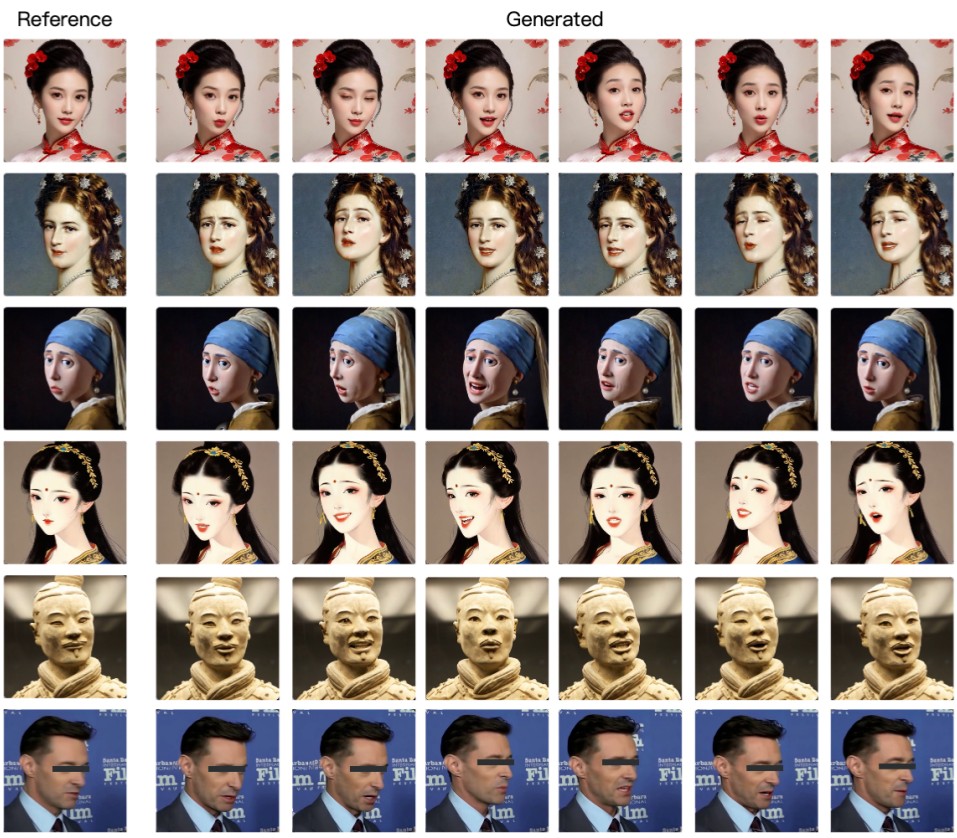

Figure 6: Visualization of videos generated by Loopy in different scenarios.

### 3.5.3 VISUAL RESULTS ANALYSIS

We provide visual analysis for the openset scenarios in Figure 6. Loopy demonstrates satisfactory synthesis results in ID preservation, motion amplitude, and image quality and also performs well with various styles of images. More video samples are available in the supplementary materials.

## 4 CONCLUSION

In this paper, we propose Loopy, an end-to-end audio-driven portrait video generation framework that can generate vivid talking portrait videos without requiring any motion constraints. Specifically, we propose the inter/intra-clip temporal module and the audio-to-latents module, which better utilize and model long-term motion information and the correlation between audio and portrait motion from temporal and audio perspectives, respectively. Extensive experiments validate the effectiveness of our method, demonstrating significant improvements in temporal stability, motion diversity, and overall video synthesis quality over existing methods.

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

## A DATASET

**Data Collection.** Datasets that simultaneously possess diversity, high visual quality, and high audio-lip sync accuracy are very scarce. We expanded our training dataset using other data sources intended for general tasks. Our training data includes the public talking head dataset HDTF (Zhang et al., 2021) and other sources from which talking head data can be obtained through post-processing, including data sources listed in OpenVid (Nan et al., 2024) and VFHQ (Xie et al., 2022), and online video platforms such as Pexels. We referred to the data review processes of previous methods (Tian et al., 2024; Xu et al., 2024a; Chen et al., 2024; Xu et al., 2024b) and designed a rigorous data usage review process. To ensure compliance with privacy standards and platform terms of use, collectors thoroughly review the videos to remove any personally identifiable information and confirm adherence to data source privacy requirements. To ensure the video does not contain irrelevant content, we finally filtered out data unrelated to talking heads using the visual language model, specifically excluding captions unrelated to humans, talking, and singing. Once the raw videos are obtained, we convert them to images containing only RGB information and further perform image post-processing and VAE feature extraction. This process transforms the videos into high-dimensional feature representations, which are stored separately and are only accessible to verified researchers for research purposes, ensuring that they will not be redistributed.

**Data Processing.** The duration of the whole dataset for Loopy and recent related works is demonstrated in the Table 4. For the data processing, we first used face landmark detection to filter out videos where the face center point movement was too large (considered as camera cuts) and removed videos with multiple people. Videos longer than 20 seconds were split into multiple segments, while segments shorter than 2 seconds were discarded. We also filtered out videos where any head pose rotation angle exceeded 45 degrees (including side profiles). After the initial filtering, we removed videos with audio that had a SyncNet video-level sync confidence below 2.5. Finally, videos smaller than 150KB were discarded. For raw videos that do not contain audio, they will be used in the first stage training. Videos selected for testing are excluded from the training split. Regarding the composition of the original training data, 23% of the videos had a maximum edge length below 512 pixels, 63% were between 512 and 1024 pixels, and 14% were above 1024 pixels. In terms of data types, 76% were talking or interview formats, 19% were singing, and 5% were direct-to-camera speeches or talk show formats.

Table 4: Statistics of dataset scale of different methods.

| Method | Total Hours |
|---|---|
| EMO (Tian et al., 2024) | 300 |
| Hallo (Xu et al., 2024a) | 109 |
| EchoMimic (Chen et al., 2024) | 540 |
| VExpress (Wang et al., 2024) | 300 |
| Loopy | 174 |

## B ADDITIONAL EXPERIMENTS DETAILS

### B.1 METRICS CALCULATION

We emoloy several objective metircs to access our Loopy. In this section, we describe their detailed calculation processes.

**IQA.** We employ Q-Align (Wu et al., 2023) to assess the visual quality of generated videos. It teaches LLMs with text-defined rating levels and achieves state-of-the-art performance on image quality assessment.

**Smo.** To assess video motion stability, we employ the smooth metric from the widely used video generation evaluation tool VBench (Huang et al., 2024). Given an input video, it returns a smooth score with value less than 1. In general, the higher the value, the smoother the motion of the video. It should be noted that for nearly static videos, smooth score still assigns a relatively high score (as shown in Tab 3, Vexpress), so we mainly use this metric for reference. A low smooth score can still indicates poor video generation stability.

**Glo.** Smooth metric identifies videos with discontinuous motion but fails to assess the motion diversiety, which is an important concren in talking head videos. Hence, we propose Glo to evaluate the extent of head movement in facial videos. It calculates the variance of the nose landmark positions detected by DWPose (Yang et al., 2023) over the video sequence to reflect the intensity of head movements.

$$Glo = \frac{\Sigma(x_i^n - \hat{x_n})^2 + \Sigma(y_i^n - \hat{y^n})^2}{t} \times s, i = 1, 2, ..., t \tag{3}$$

where $t$ is the frames number, and $x^n, y^n \in [0, 1]$ denote the coordinates of the nose landmark in the horizontal and vertical directions, respectively. $\hat{x^n}$ and $\hat{y^n}$ represent the average coordinates of the nose landmark within the clip in the horizontal and vertical directions, respectively. $s$ is a scaling factor, which is set to 1000 here. The final Exp metric is obtained by averaging the Exp values across all videos.

**Exp.** In addition to the absolute displacement of head, the richness of facial expressions also plays a crucial role in generation quality. An excellent talking head generation method produces expressive talking videos with diverse facial expressions and micro-expressions, rather than just lip movements. To assess this, We introduced the Exp metric. Given nose landmark as base, we first compute the expression representation for each frame.

$$e_i = (x_i^o - x_i^n) + (y_i^o - y_i^n) \tag{4}$$

where the superscript $n, o$ denote the nose and other facial landmarks (excluding keypoints below the nose that are strongly related to lip movements). We then calculate Exp to represent the frequency of expression changes.

$$Exp = \frac{\Sigma(d_{e_i} - \hat{d_{e_i}})^2}{t} \times s, t = 1, 2, ..., t - 1 \tag{5}$$

where $d_{e_i}$ is $e_{i+i} - e_i$. $\hat{d_{e_i}}$ is the average value within the video clip. $s$ is a scaling factor, which is set to 1000 here. The final Exp metric is obtained by averaging the Exp values across all videos.

**The calculations for Glo and Exp are also used to obtain the head movement variance and facial expression variance in the audio-to-latents module.** Notably, we use these variance features solely as auxiliary conditions during training and do not treat them as supervision signals.

**DGlo** and **DExp.** For the test set with ground truth, we also report DGlo (Diff-Glo) and DExp (Diff-Exp) to analyze the differences in Glo and Exp between each generated (subscript g) video and the ground truth (subscript gt) video. The final metric is the average value across all videos.

$$DGlo = Glo_g - Glo_{gt} \tag{6}$$

$$DExp = Exp_g - Exp_{gt} \tag{7}$$

**FVD.** We calculated the Fréchet Video Distance (FVD) (Unterthiner et al., 2019) to evaluate the quality of generated videos. FVD-R represents the evaluation with ResNet50 as the feature extraction network, and FVD-I represents the evaluation with Inception3D as the feature extraction network.

**E-FID.** We also employ E-FID following recent methods (Tian et al., 2024; Xu et al., 2024a) to evaluate the generated facial expressions. First, we extract expression parameters using face reconstruction network, and then we compute the FID of these expression parameters to quantitatively measure the divergence between the generated expressions and the ground truth.

**FID.** We employ the widely used FID to assess the generated image quality. InceptionV3 is choosed as the backbone to extract features.

**Sync-C** and **Sync-D.** We report the widely used Sync-C (confidence) and Sync-D (distance) (proposed in SyncNet (Chung & Zisserman, 2017)) to assess the lip-sync performance.

## B.2 OPENSET TEST SET DEMONSTRATION

To comprehensively evaluate our proposed Loopy, we introduce an open set in Sec 3.5.1. It consists of four categories of openset test images, real people, anime, humanoid crafts, and side faces. And Figure 7 shows several samples from it. The openset test set poses significant challenges to the model's generalization ability, yet Loopy handles them well, producing stable results and demonstrating significant advantages in the user study. Some images are from the project pages of EchoMimic (Chen et al., 2024) and Follow-Your-Emoji (Ma et al., 2024).

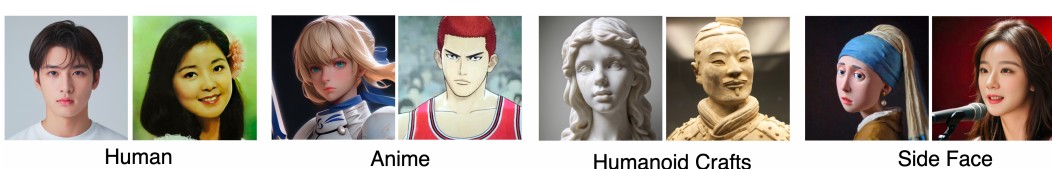

Figure 7: Visualization of test images in the openset test set

## B.3 COMPARISONS IN INFERENCE SPEED.

We also evaluated the inference speed of recent methods, testing the time taken per timestep on an unloaded A100 GPU. As shown in Table 5, most models have similar times since they are all based on the stable diffusion model, suggesting that implementation differences have a greater impact on current speeds. Notably, although Loopy uses longer motion frames and additional temporal layers, the former only needs to be computed once and can reuse features across different timesteps, while the latter operates only in the temporal dimension, minimizing the impact on inference overhead. Additionally, Loopy requires only 25 denoising steps, compared to 40 and 30 steps in the official implementations of Hallo and Echomimic, respectively. This provides a significant speed advantage.

Table 5: Comparisions of the infernece speed.

| Method | EchoMimic | Hallo | VExpress | Loopy |
|---|---|---|---|---|
| Time(s) | 0.501 | 0.642 | 0.380 | 0.758 |
| RTF | 30.8 | 40.5 | 33.4 | 43.1 |

Table 6: Comparison of denoising and video generation costs.

| Method | Denoising Cost | 15s-Video Cost | RTF |
|---|---|---|---|
| w.o.TSM | 18s | 628s | 41.8 |
| Full Model | 19s | 647s | 43.1 (+3%) |

**About the impact of TSM on overall computational efficiency.** The introduction of TSM may result in increased time consumption in two areas: (1) The VAE computation before the denoising network increases from 4 frames to 20 frames. However, this time can be omitted because the first segment uses zero-initialized motion frames, and subsequent segments use VAE latents predicted by the previous denoising net, resulting in no additional computation compared to the baseline. (2) In the denoising network, the temporal attention increases from 12+4 to 12+20.

In Table 6, we present the time comparison for generating 12 frames in 25 denoising steps, as well as the time consumption and corresponding RTF (real-time factor) for generating a 15-second video. The experiments were conducted on an unloaded A100. The introduction of TSM only increases the RTF by 3%, which we believe is acceptable given the improvement.

**Discussion on the Potential for Real-Time Inference.** Loopy takes 18 seconds to complete 25 steps of denoising to generate 12 frames on an A100 GPU, resulting in a real-time factor (RTF) of 43.1 for a 25 FPS output. By leveraging computational parallelism, we can offload audio and reference CFG inference, as well as VAE, to other GPUs for an estimated 3x speedup. Additionally, using few-step

distillation methods to reduce the steps to within 4 (estimated 6x speedup), we could potentially achieve a computation speed of around 2 seconds. Combining this with acceleration techniques like TensorRT and caching mechanisms, or using advanced GPUs like the H100 or slightly lowering the frame rate, real-time performance might be achievable. Although feasible, actual implementation may not be smooth and poses many challenges that we aim to address in future work.

## C  ADDITIONAL SUPPLEMENTARY EXPERIMENTS AND IMPLEMENTATION DETAILS

**Comparison with the Simple Baseline.** To further validate our method's effectiveness, we trained a simple baseline on the collected dataset using our pre-trained stage-1 weights. This baseline removes components like audio-to-latents, inter/intra- clip temporal layers, and temporal segment module, making it similar to EchoMimic, except for spatial condition modules like the landmark encoder and weighted loss. As shown in the Table 8, the simple baseline performs significantly worse than the full model, with substantial declines in most metrics. This, combined with the comparative experiments, clearly demonstrates that the improvements in our method primarily stem from the proposed modules.

**About the motion frame during inference.** During inference, if motion frames need to be filled, this is accomplished by duplicating the reference image multiple times. However, during training, motion frames are almost never identical. Therefore, for the first generated segment, we found that if zero initialization is not used, there is approximately a 20% chance (based on rough estimates) that the generated result will exhibit varying degrees of color difference compared to the original image.

Table 7: Comparison of different ref-CFG and audio-CFG settings.

| Ref-CFG | Audio-CFG | Sync-C | Sync-D | FVD-R |
|---|---|---|---|---|
| 1.0 | 1.0 | 7.072 | 8.154 | 18.662 |
| 2.0 | 5.0 | 8.500 | 6.981 | 16.012 |
| 2.5 | 5.0 | 8.499 | 6.976 | 14.788 |
| 3.0 | 5.0 | 8.501 | 6.983 | 14.439 |
| 3.5 | 5.0 | 8.369 | 7.254 | 14.652 |
| 3.0 | 3.0 | 8.063 | 7.337 | 14.906 |
| 3.0 | 4.0 | 8.334 | 7.193 | 14.431 |
| 3.0 | 5.0 | 8.501 | 6.983 | 14.439 |
| 3.0 | 6.0 | 8.524 | 6.957 | 15.169 |

Table 8: Comparison with the simple baseline. The metrics marked with * indicate results on the openset test set (the first 3 columns), while the others are results on the HDTF test set.

| Method | IQA* | Sync-C* | Sync-D* | IQA | Sync-C | Sync-D | FVD-R | FID | EFID |
|---|---|---|---|---|---|---|---|---|---|
| Simple Baseline | 4.33 | 5.75 | 8.17 | 3.88 | 7.95 | 7.38 | 16.86 | 21.15 | 1.68 |
| Full Model | 4.51 | 6.30 | 7.75 | 4.01 | 8.57 | 6.81 | 10.44 | 18.02 | 1.36 |

## D  LIMITATION OF CURRENT MODEL

We provided videos where Loopy performs poorly, along with an analysis, in the videos provided on our project homepage. Regarding input images, Loopy struggles with images containing large proportions of upper limbs, blurry or occluded human figures, animals, and multiple people, often resulting in incongruous or uncanny valley effects. Regarding input audio, Loopy performs poorly with long periods of silence, chaotic background noise, and extended high-pitched sounds. Overall, Loopy's synthesis quality suffers from long-tail effects, showing robustness issues and awkward movements for uncommon input types.

# E  ETHICAL RISKS DISCUSSION

Loopy possesses realistic portrait video generation capabilities, and the current experimental results are intended solely for academic research. To prevent illegal misuse if this technology is made publicly available, we think the following measures can be taken: (1) Add prominent watermarks to all generated results to indicate they are produced by AIGC algorithms; (2) Use filtering algorithms to review and intercept inappropriate, vulgar, or malicious input audio and generated video content; (3) Embed watermarks for traceability. For the ethical risks associated with data usage, we provide detailed explanations in the Dataset Section A and conduct necessary reviews during the usage process to avoid these risks.

