# OpenReview forum: "Loopy: Taming Audio-Driven Portrait Avatar with Long-Term Motion Dependency"
_ICLR.cc/2025/Conference — ICLR 2025 Oral_

### Official Review · Reviewer_eXJV · 2024-10-30

**Soundness:** 4
**Presentation:** 4
**Contribution:** 3
**Rating:** 8
**Confidence:** 4

**Summary:**

This paper introduces an audio2video model for co-speech human portrait video synthesis. A novel temporal module is proposed to enable natural movement generation. A joint audio, movement, and expression latent space is learned to achieve better head pose and facial expression control from speech. Experiments and demonstrations show better performance and more realistic results.

**Strengths:**

1. The results are good.
2. The introduction of two modules (Temporal and Audio) is reasonable and interesting. Ablation study supports the benefits of these modules.

**Weaknesses:**

1. Lack of ablation of stand-alone intra- / inter-temporal model. Is both of them necessary or only the inter-clip temporal layer is enough?
2.  The functionality of the Temporal Segment Model is unclear. Is it for capturing the appearance of the character under different expressions? If so, why (L478) longer motion frames lead to worse results?
3. Similar to the above issue. I watched the video samples of the ablated model. Seems to me the ablation of either part leads to similar degradations — lack of head pose variance and subtle expression. This makes me unclear about the different roles of the two proposed modules.

**Questions:**

1. During inference what if motion frames are provided? How would they influence the results?
2. Can the overall head motion be controlled?
3. (L291) Is there any analysis of the strong correlation between the head movement and expression variances? Can the type of expression be controlled?

**Details Of Ethics Concerns:**

The model can be used for deepfake kind of misleading video generation.

---

> ### Author Response · Authors · 2024-11-22
> **Author Response to Reviewer eXJV (Part 1/2)**
>
> We are grateful for your positive review and valuable comments, and we hope our response fully resolves your concerns.
>
> > Q1: About the ablation of inter/intra- clip layers
>
>   First, we apologize for not clearly explaining this in the paper. These results are actually already included, corresponding to "w/o inter-clip temp." in Line 463 and “1 temp.+ 20 MF” in Line 466. After removing the inter-clip temporal layer, to retain the information of motion frames, the remaining single temporal layer becomes the setting for "inter-clip temporal layer only." The two results mentioned above correspond to different numbers of motion frames (4 and 20). It can be observed that regardless of the setting, there is a significant performance drop compared to the full model.
>   For the setting where the inter-clip temporal layer is completely removed and motion frames are not retained, the synthesized long videos exhibit noticeable misalignment every 12 frames, rendering the results unusable. Therefore, we did not evaluate this setting.
>
> > Q2: About TSM functionality and worse results with longer motion frames.
>
>   From Lines 466-469, it can be observed that applying the dual temporal layer improves overall performance as the number of motion frames increases (by adjusting TSM parameters for longer coverage). This indicates that the TSM helps establish longer-term motion dependencies, enhancing final performance.
>    In contrast, Line 478 states that increasing the number of motion frames leads to performance degradation when using a single temporal layer.
>   This is because, with a single temporal layer, the temporal information needs to handle 20 frames of non-noisy latent features from the last clip and 12 frames of latent features from the current timestep. When the former dominates, it affects the temporal modeling of the noisy latents in the current clip, thereby impacting the generation task. In contrast, the dual temporal layer can distinguish between these two scenarios: one temporal layer models the inter-clip temporal information dominated by non-noisy latent features from the last clip, while the other models the intra-clip temporal information dominated by the noisy latents of the current clip. This distinction enhances performance.
>
> > Q3: About different roles of the two proposed modules.
>
>   The proposed modules directly impact motion, so intuitively, removing any module will lead to a decline in motion modeling capability and overall expressiveness.
>   1. From the supplementary materials, it can be observed that removing the audio2latents module results in a more noticeable decline in eyebrow and eye movements (independent movements rather than overall motion), which are the parts that its auxiliary spatial condition can control.
>   2. We provide results from the challenging test set CelebV-HQ in [video-abalation](https://loopyavataranony.github.io/video/rebuttal_video_aba.mp4), where the model without the proposed modules struggles to handle, especially without the inter/intra-clip temporal layers, better demonstrating their functionality. It can be observed that models without the inter/intra-clip temporal layer are more prone to random movements and temporal inconsistencies, while models without the Audio-to-Latents module tend to produce more expressionless results.
>   Overall, both proposed modules enhance motion modeling, but with different focuses: the inter/intra-clip temporal layer improves temporal naturalness and continuity, while the Audio2Latents module enhances subtle expression generation. This distinction is more evident in complex examples.
>
> > Q4: About the motion frames setting during inference
>
>   During inference, if motion frames need to be filled, this is achieved by duplicating the reference image multiple times. However, during training, motion frames are almost never exactly the same. Therefore, for the first generated segment, if zero initialization is not used, there is about a 20% chance (based on rough estimates) that the generated result will have varying degrees of color difference compared to the original image. Other than this, there is almost no difference. Additionally, zero-initializing motion frames for the first segment can save one motion frame VAE encoding time (although we later found this impact to be minimal, we retained this setting).

---

> ### Author Response · Authors · 2024-11-22
> **Author Response to Reviewer eXJV (Part 2/2)**
>
> > Q5: About whether the head is controllable and movement and expression variances.
>
> Because Loopy is an end-to-end audio-only model, we cannot precisely control head movements. However, the training of audio2latents allows some influence over head movements and expressions. In [video-control_demo](https://loopyavataranony.github.io/video/rebutall_video_expmov_control.mp4), we show results of replacing audio features with specified exp and var values in the audio2latents module. As input values change, the generated results vary accordingly, similar to controlling expression types.
>
> **Summary**
>
> Based on your comments, we analyzed the implementation and effectiveness of dual temporal layers and TSM in our responses, and added a discussion on the motion frame inference setting in the revised version. Thank you again for your positive comments and valuable suggestions.

---

> > ### Comment · Reviewer_eXJV · 2024-11-26
> >
> > Thanks for the comments. I think the additional results and explanation fully validated the proposed method.  The video-control demo is interesting and promising. I suggest making a point of this in the main paper. Above all, I think this is a good paper with carefully designed and validated modules and interesting applications. I would like to raise my score.
> >
> > Thanks again for the hard work!

---

> > > ### Author Response · Authors · 2024-11-26
> > >
> > > We are very pleased that our response has addressed your concerns. We greatly appreciate your recognition of our work and the constructive suggestions you provided, which have made the paper more complete.
> > >
> > > Following your suggestions, we have added a description in Line 308 describing the audio-to-latents module can also influence motion generation through the input expressions and movements. Due to space limitations, we will include relevant explanatory videos (i.e., rebuttal videos) on the project page so that readers can more intuitively observe the differences in controlled movements.

---

### Official Review · Reviewer_bAAG · 2024-11-02

**Soundness:** 4
**Presentation:** 3
**Contribution:** 3
**Rating:** 8
**Confidence:** 4

**Summary:**

This paper proposes an end-to-end audio-driven portrait video generation method. This method introduces an inter- and intra-clip temporal module and an audio-to-latent module to establish long-term natural correlations between audio and portrait movements. Many lifelike and impressive results are presented.

**Strengths:**

1. The motivation is clear. The authors focus on the weak correlation between audio and portrait motion in end-to-end audio-driven methods.
2. Overall, this paper is easy to follow. The proposed TSM module is technically sound in its design, and the experimental validation is effective.
3. Many synchronized and vivid portrait videos are generated.

**Weaknesses:**

1. In the A2L module, the effects of Movement and Expression on the method have not been thoroughly validated. The audio inputs shown in Fig. 4 are somewhat confusing. I assume they refer to audio features from wav2vec.
2. Human expressions are closely related to many facial details, but the implementation in the paper is rather trivial.
    1) the detected landmarks are too sparse and not accurate enough (DWPose), which makes it difficult to capture a person's expression accurately.
    2) using the variance of keypoints to calculate head movement and expression changes presents several practical issues,
such as the entanglement of head movement and camera movement. Why not use FLAME coefficients or results from other emotion estimation methods?
3. The TSM module needs a deeper discussion on its impact on overall computational efficiency.
4. In Tables 1 and 2, the methods perform worse than others on some metrics, especially those related to Glo and Exp. The authors do not provide detailed analysis or discussion on this.
5. The paper has several writing issues. Some symbols and abbreviations are introduced without explanation, such as TSM in Fig. 2. Additionally, some text in the figures is too small to read, such as "other computational layers" in Fig. 3. The main paper does not reference Table 2. There are also some typos, such as in Line 302, where there is an error with punctuation.
6. The paper does not include a discussion of the limitations of the proposed method.

**Questions:**

1. Currently, end-to-end audio-driven portrait generation is typically trained on training sets of varying sizes, which is crucial for a good model. How can we reasonably evaluate the performance of the method?
2. In Table 3, the metrics for audio-visual synchronization related to Loopy w/o TSM and w/o ASL still outperform other methods. Does this indicate that the performance improvement of the method primarily comes from the self-collected data?
3. Regarding training A2L, how do head movements and expressions individually affect the results?

---

> ### Author Response · Authors · 2024-11-22
> **Author Response to Reviewer  bAAG (Part 1/2)**
>
> We are grateful for your positive review and valuable suggestions, which are also constructively helpful for our subsequent research, and we hope our response fully resolves your concerns.
>
> > Q1: About the effects of movement and expression and implementation details.
>
>  We first apologize for any lack of clarity in our previous explanation. Your understanding is correct. the term "audio" here refers to the concatenated features from the multiscale outputs of wav2vec, as described in Line 280. For the effects of movement and expression, in Table 3, we conducted a abalation experiment by removing the audio-to-latents module, which resulted in a noticeable decline in performance. Additionally, in [video-abalation](https://loopyavataranony.github.io/video/rebuttal_video_aba.mp4) and the supplementary materials, it can be seen that the audio-to-latents module enhances the expressiveness of movements, such as the eyebrows and eyes. In [video-control_demo](https://loopyavataranony.github.io/video/rebutall_video_expmov_control.mp4), we also demonstrate the ability to control movement and expression independently, illustrating their relationship with portrait motion. Based on this, strong movement-related signals can help the model better learn the relationship between audio and portrait movement.
>
> > Q2: Regarding expressions and movement detection.
>
> Thank you for your excellent suggestion. First, we provide some [video-detection_demo](https://loopyavataranony.github.io/video/rebuttal_video_expmov_det.mp4) to demonstrate that the current landmark-based calculations of var and exp can capture most of the variations in facial expressions and head movements. Based on these observations, we initially chose the landmark-based detection method in our experiments, which currently generates satisfactory results. Second, it is evident that the current landmark detection does have some motion coupling issues. FLAME-based detection can bring intuitive improvements. We are considering incorporating training strategies with detection confidence scores, which might further enhance the role of auxiliary spatial conditions during training. We will investigate this further in future work. Thank you.
>
> > Q3: About the computational efficiency after introducing TSM
>
> The introduction of TSM may result in increased time consumption in two areas: (1) The VAE computation before the denoising network increases from 4 frames to 20 frames. However, this time can be omitted because the first segment uses zero-initialized motion frames, and subsequent segments use VAE latents predicted by the previous denoising net, resulting in no additional computation compared to the baseline. (2) In the denoising network, the temporal attention increases from 12+4 to 12+20.
>
>   Below, we present the time comparison for generating 12 frames in 25 denoising steps, as well as the time consumption and corresponding RTF (real-time factor) for generating a 15-second video. The experiments were conducted on an unloaded A100. The introduction of TSM only increases the RTF by 3%, which we believe is acceptable given the improvement.
> | Method | Denoising Cost | 15s-Video Cost | RTF |
> | --- | --- | --- | --- |
> | w.o.TSM | 18s | 628s | 41.8 |
> | Full Model | 19s | 647s | 43.1(+3%) |
>
> > Q4: About more analysis on the performance in Table 1 and 2.
>
> In Tables 1 and 2, Loopy achieves the top performance on most metrics. Although it ranks second on some lipsync and EFID metrics, the differences are minimal (lipsync and expression matching evaluations are inherently challenging and subject to some degree of fluctuation or error). As for GLO and EXP, which represent the absolute magnitude of movements, bigger values are not necessarily better. These metrics are provided to illustrate the preference differences in movement magnitude among different methods, with some methods tending towards stillness to avoid errors. The metrics DGLO and DEXP represent the deviations of GLO and EXP from the corresponding values in the ground truth (GT) videos, and Loopy still leads in most cases.
>   Since objective metrics may have some inaccuracies, we also include subjective evaluations in Figure 5 to validate Loopy's superiority across multiple dimensions. The results show that Loopy has an overwhelming advantage.
>
> > Q5: About the writing issues.
>
> Thank you very much for pointing out these issues. We have made the revisions in the revised version, reviewed the writing again, and corrected some inappropriate uses of punctuation, such as ":".

---

> ### Author Response · Authors · 2024-11-22
> **Author Response to Reviewer bAAG (Part 2/2)**
>
> > Q6: About the discussion of the limitations of the proposed method.
>
>   Thank you for your reminder. Our intention in using an open-set test set is to explore the capability boundaries of the Loopy model. In [video-limitation](https://loopyavataranony.github.io/video/rebuttal_video_limitation.mp4)  we provide an analysis of the current method's limitations.
>   Regarding input images, Loopy struggles with images containing large proportions of upper limbs, blurry or occluded human figures, animals, and multiple people, often resulting in incongruous or uncanny valley effects.
>   Regarding input audio, Loopy performs poorly with long periods of silence, chaotic background noise, and extended high-pitched sounds.
>   Overall, Loopy's synthesis quality suffers from long-tail effects, showing robustness issues and awkward movements for uncommon input types.
>
> > Q7: About evaluating the method's performance reasonably
>
>   Yes, this is indeed a perplexing issue for us as well. We have strived to provide a fair comparison by training our model on HDTF. Even with only 1/10 to 1/20 of the training data compared to other methods, Loopy still achieves leading performance on several key metrics. On the collected dataset, Loopy also shows significant improvements through comparative experiments and provided videos, demonstrating the effectiveness of the proposed modules both qualitatively and quantitatively.
>   From an evaluation perspective, misaligned datasets pose a challenging problem for evaluation. Retraining methods on large datasets involves significant time and setup costs, especially for diffusion models. We currently believe that demonstrating module effectiveness through comparative experiments and showing qualitative and quantitative improvements in final results to achieve state-of-the-art performance is a suitable approach.
>
> > Q8: About the relationship between improvement and self-collected data
>
>   Based on the considerations from the previous response, we provide a simple baseline trained on the collected dataset. This baseline removes components such as audio2latents, dual temporal layers, and TSM, making it very close to echomimic, except for spatial condition modules like the landmark encoder and weighted loss, which are functional components or independent optimization points. This comparison better illustrates the effectiveness of our method. The significant performance drop observed when all proposed modules are removed underscores their importance.
> | Method | openset IQA | openset SyncC | openset SyncD | HDTF IQA | HDTF SyncC | HDTF SyncD | HDTF FVD | HDTF FID | HDTF EFID |
> | --------------------------- | --- | --- | --- | --- | --- | --- | --- | --- | --- |
> | Simple Baseline | 4.33 | 5.75 | 8.17 | 3.88 | 7.95 | 7.38 | 16.86 | 21.15 | 1.68 |
> | Full Model | 4.51 | 6.30 | 7.75 | 4.01 | 8.57 | 6.81 | 10.44 | 18.02 | 1.36 |
>
>   Regarding the audio-visual synchronization metrics, our method leads even without the proprietary dataset (HDTF only), outperforming other methods that use 10-20 times more training data. This indicates that our performance advantage is not solely due to the amount of data. In fact, the HDTF training set includes very high-quality speech videos, and we found that audio-visual synchronization is relatively easy to learn. The increase in data volume primarily enhances the overall synthesis quality, such as image quality and the learning of complex motion patterns, which also benefits from the efficient utilization of data by the proposed modules.
>
>   The poorer performance of the compared methods could be due to various reasons. For example, echomimic exhibits large movements but poor lipsync, while hallo shows the opposite. It is possible that these methods did not effectively leverage the long-term motion dependencies in the data, and the introduction of spatial conditions caused motion pattern misalignment between training and testing, resulting in imbalanced model performance and suboptimal overall results.
>
> > Q9: About the independent control of head movements and expressions
>
>   Certainly. In [video-control](https://loopyavataranony.github.io/video/rebutall_video_expmov_control.mp4), we provide the results of replacing the audio features input with specified exp values and var values in the audio2latents module. As the input values change, the generated results also exhibit corresponding variations.
>
>
> **Summary**
>
> Based on your comments, we have added an analysis of the computational efficiency impact of TSM, addressed the limitations of loopy, and provided a simple baseline to highlight our module's effectiveness. We appreciate the reviewer's suggestions on head movement and expression detection using FLAME, which will aid our further research. Thank you for your positive rating and valuable feedback.

---

> > ### Comment · Reviewer_bAAG · 2024-11-26
> >
> > I would like to thank the authors for their response. They have satisfactorily answered my questions. Therefore, I will keep my previous score and vote to accept the paper.

---

> > > ### Author Response · Authors · 2024-11-26
> > >
> > > Thank you very much for recognizing our work, which is truly encouraging. We also appreciate your valuable suggestions, as they have further improved the quality of our work and provided us with great insights.

---

### Official Review · Reviewer_2GHy · 2024-11-04

**Soundness:** 4
**Presentation:** 3
**Contribution:** 4
**Rating:** 8
**Confidence:** 4

**Summary:**

The paper presents Loopy, an innovative audio-driven diffusion model for generating portrait videos that addresses limitations in current methods related to motion naturalness and dependency on auxiliary spatial signals. Existing approaches often compromise the natural freedom of movement by using preset spatial constraints like movement regions or face locators to stabilize motion, leading to repetitive and less dynamic results.

Loopy stands out by adopting an end-to-end audio-only conditioning framework, leveraging two main components: 1. Inter- and Intra-clip Temporal Modules: These modules are designed to extend the model’s temporal receptive field, enabling it to utilize long-term motion dependencies and generate consistent, natural motion across video frames without external movement constraints; 2. Audio-to-Latents Module: This module enhances the correlation between audio input and portrait motion by converting audio features and motion-related characteristics into latent space representations that guide the synthesis process.

Experiments show that Loopy outperforms existing methods, generating lifelike and stable videos with natural facial expressions.

**Strengths:**

1. The paper introduces an end-to-end audio-only conditioned video diffusion model, which moves beyond traditional methods that rely on spatial constraints for motion stabilization.

2. The proposed novel modules like inter- and intra-clip temporal modules and audio-to-latents module are well-designed, resulting in more natural and consistent portrait movements and leading to better synchronization and more expressive facial movements.

3. The paper includes extensive experiments that demonstrate Loopy’s superiority over other audio-driven portrait diffusion models both quantitatively and qualitatively, with evidence of more lifelike and stable video outputs in the supplemental website.

4. The paper is well-written, the proposed components and architecture are described clearly.

**Weaknesses:**

1. While the audio-to-latents module improves the audio-motion correlation, there is no mention of how different audio characteristics (e.g., background noise, varying loudness) might impact the model’s performance, which could be critical for real-world applications.

2. The paper lacks a detailed analysis of potential failure modes or scenarios where Loopy may struggle. Highlighting these cases would provide a more balanced view of the model's robustness and limitations.

**Questions:**

1. Are there specific cases where Loopy struggles to maintain natural motion or facial expressions? An analysis of these limitations would provide a more complete understanding of the model’s strengths and weaknesses.

2. In the experiments section, the baseline models compared with Loopy were not trained using the collected dataset. It would be helpful to see how these baseline models perform when trained on the same dataset. This could further validate the effectiveness of the proposed modules and confirm that the performance gains are due to the model’s design, rather than advantages inherent to the dataset itself.

3. Will the collected dataset be made publically available?

---

> ### Author Response · Authors · 2024-11-22
> **Author Response to Reviewer 2GHy**
>
> We are grateful for your positive review and valuable comments, and we hope our response fully resolves your concerns.
>
> > Q1: Regarding the impact of different audio characteristics (e.g., background noise, varying loudness) on the model’s performance
>
> Thank you for your meaningful suggestion. Below, we provide an analysis of the model's performance with audio inputs of varying disturbance levels (adding different Gaussian noise and applying random volume scaling). Due to time constraints, we tested the first 20 examples from the HDTF test set. It can be observed that even with significant background noise (which is already quite disruptive for humans, 0.02 noise), Loopy still maintains good audio-visual synchronization and performs comparably to existing methods under normal input conditions. However, for noise that masks human voices, Loopy's performance noticeably declines. In terms of loudness disturbances, Loopy is relatively robust and can still achieve good results. Therefore, Loopy demonstrates good robustness to relatively common abnormal samples. In our response to Q2, we have also provided videos and further analysis of the limitations.
> | Augmentation_type | sync-c |  FVD-R  |
> |:-----------------:|:------:|:-------:|
> |     0.01 noise    | 7.1179 | 14.932  |
> |     0.02 noise    |  5.698 | 15.428  |
> |     0.05 noise    |  3.529 | 19.792  |
> |     0.1 scale     |  8.230 | 14.351  |
> |     0.25 scale    |  8.224 | 14.964  |
> |     0.5 scale     |  8.104 | 14.559  |
> |     EchoMimic (Normal)   |  5.820 | 22.650  |
> |     Loopy (Normal)     |  8.501 | 14.439  |
>
> > Q2: Regarding the imitations of the proposed method and failure cases.
>
> Thank you for your reminder. Our intention in using an open-set test set is to explore the capability boundaries of the Loopy model. In [video-limitation](https://loopyavataranony.github.io/video/rebuttal_video_limitation.mp4) we provide an analysis of the current method's limitations.
> Regarding input images, Loopy struggles with images containing large proportions of upper limbs, blurry or occluded human figures, animals, and multiple people, often resulting in incongruous or uncanny valley effects.
> Regarding input audio, Loopy performs poorly with long periods of silence, chaotic background noise, and extended high-pitched sounds.
> Overall, Loopy's synthesis quality suffers from long-tail effects, showing robustness issues and awkward movements for uncommon input types.
>
> > Q3: Regarding the influence of the training data and other baselines trained on our collected dataset.
>
> To validate our method's effectiveness, we demonstrated its impact in two ways:
>
> (1) In Table 3, results trained on HDTF-only show our method outperforms other baselines on key metrics like audio-visual synchronization (sync-c/sync-d) and overall video quality (FVD), even without additional collected data.
>
> (2) The ablation analysis in Table 3 and the supplementary videos highlights the improvements of the proposed modules from both quantitative and qualitative perspectives.
>
> Training on the same dataset is a valuable suggestion. Due to limited time and different stage-1/2 training parameters of different methods, we trained a simple baseline on the collected dataset using our pre-trained stage-1 weights. This baseline removes components like audio2latents, dual temporal layers, and TSM, making it similar to echomimic, except for spatial condition modules like the landmark encoder and weighted loss. As shown in the table below, the simple baseline performs significantly worse than the full model, with substantial declines in most metrics, even falling short of the full model trained only on HDTF. This, combined with comparative experiments, clearly demonstrates that the improvements in our method primarily stem from the proposed modules.
> | Method | Openset IQA | Openset Sync-C | Openset Sync-D | HDTF IQA | HDTF Sync-C | HDTF Sync-D | HDTF FVD-R | HDTF FID | HDTF EFID |
> | --------------------------- | --- | --- | --- | --- | --- | --- | --- | --- | --- |
> | Simple Baseline | 4.33 | 5.75 | 8.17 | 3.88 | 7.95 | 7.38 | 16.86 | 21.15 | 1.68 |
> | Full Model | 4.51 | 6.30 | 7.75 | 4.01 | 8.57 | 6.81 | 10.44 | 18.02 | 1.36 |
>
> > Q4: Regarding the public availability of the dataset
>
> At this time, we are indeed unable to provide a definite plan for making the dataset publicly available, as this decision requires further consideration and internal data review processes. We have included data filtering details and data composition in the revised version, and we hope this will be helpful.
>
> **Summary**
>
> Thank you for your valuable suggestions. In the revised version, we have added discussions on the limitations of Loopy and comparisons with the simple baseline. We believe this will make the paper more complete. Thank you again for your suggestions and positive feedback on our work.

---

> > ### Comment · Reviewer_2GHy · 2024-12-02
> >
> > Thanks the authors for the responses, most of my concerns have been addressed. I will maintain my score for acceptance.

---

### Official Review · Reviewer_tNAg · 2024-11-11

**Soundness:** 4
**Presentation:** 4
**Contribution:** 4
**Rating:** 8
**Confidence:** 5

**Summary:**

The paper proposes an audio-only conditioned video diffusion model. The model consists of three key components: an inter- and intra-clip temporal module, and an audio-to-latents module. These modules are designed to facilitate long-term movement modeling, enhancing the correlation between audio and motion. During inference, a single reference image as well as the audio is sent as input to autoregressively generate future frames window by window.

**Strengths:**

1. The proposed method is solid, with enough technical contributions to address the long-term dependency between motions and audio conditions.

2. The experiment results are strong enough compared to prior works and baselines, in particular on FVD metrics and DExp metrics.

3. Both qualitative results and the demos shown in the supplementary webpage are appealing and convincing enough, where the long-term dependencies and correlations between audio and portrait motions are consistently maintained.

4. Overall, the paper is well-written and easy to follow, albeit having many technical details.

5. The human study results clearly show that the proposed method perceptually outperforms other baselines and prior arts.

**Weaknesses:**

1. For audio-to-latent module, why replacing it with cross-attention module leads to largest performance drop as seen in Table 3. What are missing from cross-attention that makes it fail to perform as good.

2. During inference, audio ratio and ref ratio are manually set for classifier guidance, an ablation study is suggested to their impact on the final quality of generated video to have some insights about this weighting scheme.

3. Could the proposed method be further optimized and adapted to real-time settings, where the audio is being played and video follows interactively?

4. What are limitations of the proposed method and what could be improved? Are there failure cases where the generated motions cannot follow the audio closely?

**Questions:**

See weaknesses above.

---

> ### Author Response · Authors · 2024-11-22
> **Author Response to Reviewer tNAg**
>
> We are grateful for your positive review and valuable comments, and we hope our response fully resolves your concerns.
>
> > Q1: Regarding the performance degradation caused by the removal of the audio-to-latent module.
>
> First, we apologize for any lack of clarity in our previous explanation. In Table 3, "w/o A2L" refers to the removal of the A2L module while still retaining the injection of audio features into the denoising network via cross-attention ( indicated by the audio attention layer in Figure 2 and Line 283). Removing the A2L module means the model retains only the original audio-video pixel mapping relationship, reducing its capacity to model audio-portrait relationships compared to the full model. Although the A2L module highlights head and facial expression changes, the audio2latents can still provide additional audio control information about the portrait. Mapping to latents also facilitates better learning of the relationship between audio and movement, such as lip movement, which can be considered a combination of several basic features (similar to blendshapes). Consequently, this leads to performance degradation, with the most significant impact observed in the audio-related metrics, sync-c and sync-d, especially on the challenging open-set test set.
>
> > Q2: Regarding the weighting scheme of audio ratio and ref ratio
>
> Thank you for the reminder. Our initial parameter settings were based on visual observations of a few cases. Below, we provide the experimental results where we fix either the audio ratio or the reference ratio and vary the other parameter. Due to time constraints, we tested the first 20 examples from the HDTF test set for lipsync and FVD. It can be observed that increasing the audio ratio gradually enhances the accuracy of lip synchronization, reflecting the relevance of the audio, until it eventually saturates. On the other hand, increasing the reference ratio improves the overall video synthesis quality, but an excessively high reference ratio can actually degrade the quality. This may be because the reference image not only affects identity preservation but also influences the degree to which the generated result follows the reference image. An excessively high cfg can lead to reduced motion amplitude and even color changes. A reference ratio of around 3 and an audio ratio of around 5 appear to be a well-balanced choice.
> | RefCFG | AudioCFG | Sync-C | Sync-D | FVD-R |
> | --- | --- | --- | --- | --- |
> | 1.0 | 1.0 | 7.072 | 8.154 | 18.662 |
> | 2.0 | 5.0 | 8.500 | 6.981 | 16.012 |
> | 2.5 | 5.0 | 8.499 | 6.976 | 14.788 |
> | 3.0 | 5.0 | 8.501 | 6.983 | 14.439 |
> | 3.5 | 5.0 | 8.369 | 7.254 | 14.652 |
> | 3.0 | 3.0 | 8.063 | 7.337 | 14.906 |
> | 3.0 | 4.0 | 8.334 | 7.193 | 14.431 |
> | 3.0 | 5.0 | 8.501 | 6.983 | 14.439 |
> | 3.0 | 6.0 | 8.524 | 6.957 | 15.169 |
>
> > Q3: Regarding further optimization and adaption to real-time settings.
>
> Very valuable suggestion. This is exactly what we aim to explore in our future work. Currently, it takes 18 seconds to complete 25 steps of denoising to generate 12 frames on an A100 GPU, resulting in a real-time factor (RTF) of 43.1 for a 25 FPS output. By leveraging computational parallelism, we can offload audio and reference CFG inference, as well as VAE, to other GPUs for an estimated 3x speedup. Additionally, using few-step distillation methods to reduce the steps to within 4 (estimated 6x speedup), we could potentially achieve a computation speed of around 2 seconds.
> Combining this with acceleration techniques like TensorRT and caching mechanisms, or using advanced GPUs like the H100 or slightly lowering the frame rate, real-time performance might be achievable. Although feasible, actual implementation may not be so smooth and poses many challenges that we aim to address in future work. Thank you.
>
> > Q4: Regarding the imitations of the proposed method and failure cases.
>
> Thank you for your reminder. Our intention in using an open-set test set is to explore the capability boundaries of the Loopy model. In [video- limitation](https://loopyavataranony.github.io/video/rebuttal_video_limitation.mp4), we provide an analysis of the current method's limitations.
> Regarding input images, Loopy struggles with images containing large proportions of upper limbs, blurry or occluded human figures, animals, and multiple people, often resulting in incongruous or uncanny valley effects.
> Regarding input audio, Loopy performs poorly with long periods of silence, chaotic background noise, and extended high-pitched sounds.
> Overall, Loopy's synthesis quality suffers from long-tail effects, showing robustness issues and awkward movements for uncommon input types.
>
> **Summary**
>
> Following your comments, we added discussions on multi-CFG impact, real-time inference potential, and method limitations in the revised appendix. We think this enhances the paper and better explains the method. Thank you again for your valuable comments and positive recommendation for our paper.

---

### Author Response · Authors · 2024-11-24
**General Response to All Reviewers**

We greatly appreciate the reviewers' positive comments on our proposed method, recognizing our contribution as good or even excellent, which is very encouraging for us. In our responses, we followed the reviewers' comments and provided additional experimental results and discussions. We believe these additions make the paper more complete and better support the effectiveness of our proposed modules.

---

**Regarding the method itself**

- We added a description of the current limitations in Appendix D and included a [video](https://loopyavataranony.github.io/video/rebuttal_video_limitation.mp4) describing these limitations on the provided project page.
- We provided more experimental results to support the effectiveness of our method, including comparisons between a simple baseline and the full model in Appendix C, and additional comparison [video](https://loopyavataranony.github.io/video/rebuttal_video_aba.mp4) on the provided project page showing the impact of the proposed modules.
- We also provided explanations and detection results ([video](https://loopyavataranony.github.io/video/rebuttal_video_expmov_det.mp4)) for expressions and movements in the audio-to-latents module, demonstrating their independent control over the generated results ([video](https://loopyavataranony.github.io/video/rebutall_video_expmov_control.mp4)), which is also included on the provided project page.
- We included discussions and experimental results on the impact of audio and reference CFG on the results in Appendix C.

**Regarding implementation details and deployment**

- We discussed the effect of TSM on inference speed, the current model's real-time factor (RTF) analysis, and potential real-time inference implementation in Appendix B.3, and the impact of motion frames during inference on the results in Appendix C.

We also revised the paper, including adjustments to the font size and module descriptions in Figures 2 and 3, and fixed the writing mistakes in the use of formulas, letters, and punctuation, making the paper easier to understand.

---

We are very grateful for all the valuable suggestions and positive recommendations from the reviewers. We believe the above additions regarding the method itself and its application implementation greatly enhance the completeness and value of the paper. We have carefully addressed the main concerns and provided detailed responses to each reviewer. We hope you will find the responses satisfactory. We would be grateful if we could hear your feedback regarding our answers to the reviews.


\
Sincerely

Authors of Paper 4292

---

### Author Response · Authors · 2024-11-25
**Dear AC and reviewers,**

Firstly, we would like to express our sincere gratitude for organizing this conference and the review process. Based on the comments from reviewers, we have added more experimental results and included more visualizations and explanations to better support our method's effectiveness and the completeness of the paper. As the discussion period is nearing its end, we look forward to receiving feedback to ensure all concerns are fully addressed and are happy to clarify any remaining points.

Thank you once again for your time and effort.

\
Sincerely

Authors of Paper 4292

---

### Meta-Review · Area_Chair_nxFp · 2024-12-20

**Metareview:**

The paper presents Loopy, an audio-driven portrait animation pipeline that demonstrates natural talking-head motion. The key innovation in Loopy against prior works is the use of diffusion-based synthesis using long-range motion frames that leads up to the currently synthesized frame, thus allowing access to the context for producing subtle, consistent, and continuous head movements. The model is trained on a very large dataset and experiments show state-of-the-art results. The provided videos show high-quality in talking head generation.

The paper received four reviews, all recommending accept. The reviewers liked the quality of the generated videos, quality of the contributions, organization of the paper, and the extensive experimental results presented demonstrating state-of-the-art results.

**Additional Comments On Reviewer Discussion:**

There were several issues pointed out by the reviewers, which were clarified with the authors during the discussion phase. There were three main areas that had reservations from the reviewers.

1. Inconsistencies in the reported performance, missing ablation studies, and failure cases (tNAg, bAAG, eXJV)
2. The baselines models being not trained on the same dataset (2GHy, bAAG)
3. Public availability of the proposed dataset (2GHy)

For 1., authors point out that removing the audio-to-latent module reduces the capacity of the model leading to inferior results. Authors also provide additional numerical results to address other performance and ablation study issues pointed out by the reviewers. Several situations where Loopy struggles to produce reasonable videos are also provided. Authors also point out that the introduction of the temporal segment module (TSM) only brings an additional 3% increase in the compute requirement.

For 2., the paper provided some limited additional results during the discussion that compare a prior method: Echomimic and show that prior method's performance is still inferior. Authors also claim (during the discussion phase) that even with only 1/10-th or 1/20-th of training data compared to other methods, Loopy continues to achieve state-of-the-art performance, although no empirical results are furnished to substantiate this.

For 3., Authors cannot commit on releasing the dataset publicly.

Overall, AC concords with the reviewers that Loopy makes solid contributions to audio-driven video synthesis and thus recommends acceptance. However, the lack of complete experiments comparing Loopy to prior models on the same dataset or comparing Loopy on standard benchmarks make the results hard to compare against in follow up papers, unless the dataset is also released. Authors should address these issues in the camera-ready. There are also many typos in the paper and a careful proofreading is suggested.

---

> ### Public Comment · ~Jianwen_Jiang1 · 2025-02-22
> **Camera Ready Revision**
>
> Thank you to the AC and reviewers for recognizing our work. We have made modifications to the camera-ready version according to the instructions.
>
> For testing, most current diffusion-based audio-driven portrait synthesis methods use self-collected datasets of similar scale from mixed sources. In addition to differences in training sets, there is no clear consensus on test set selection in most methods. To address this and provide a fairer comparison, we have included the results of the open-sourced Hallo trained on our dataset and provided the split and processing of the test set on the project page. We have also included a detailed description of the dataset.
>
> For writing, we have carefully reviewed the entire paper in the hope of further enhancing the quality of the camera-ready version.
>
> Finally, we express our sincere gratitude to the conference organizers for their efforts and thank them for the valuable feedback that has significantly improved the quality of our work.

---

### Decision · Program_Chairs · 2025-01-22

Accept (Oral)